# Investigating Differences in Personality Traits, Self-Esteem, Eating Attitudes, and Body Image among Participants in Indoor and Outdoor Fitness Activities

**DOI:** 10.3390/healthcare12010047

**Published:** 2023-12-25

**Authors:** Ioannis Tsartsapakis, Georgios Chalatzoglidis, Aglaia Zafeiroudi

**Affiliations:** 1Department of Physical Education and Sport Sciences at Serres, Aristotle University of Thessaloniki, 62100 Serres, Greece; ioantsar@phed-sr.auth.gr (I.T.); gchalatzo@phed-sr.auth.gr (G.C.); 2Department Physical Education and Sport Science, University of Thessaly, 42100 Trikala, Greece

**Keywords:** leisure, well-being, healthy lifestyle, recreation, physical exercise, psychological health

## Abstract

Physical activity environments influence physical condition, nutrition, individual and social behaviors, and emotional well-being. The aim of the study was to investigate differences in eating attitudes, self-esteem, personality traits, and body image among participants in indoor and outdoor leisure-time fitness programs. Participants included 1747 adults (882 men and 865 women) aged 34 ± 9.2 years, divided into experimental and control groups. All participants completed the Demographic Characteristics Questionnaire, the Eating Attitudes Test 26 (EAT-26), the Self-Esteem Scale (RSES), the Traits Personality Questionnaire 5 (TPQue5), and the Appearance Scales of the Multidimensional Body–Self Relations Questionnaire (MBSRQ-AS). The results revealed statistically significant differences between groups. One-way analysis of variance (ANOVA) revealed noteworthy variations in personality trait scores between the groups (neuroticism, extraversion, and conscientiousness) as well as body image factors (self-esteem and eating attitudes). Two-way ANOVA revealed a significant interaction between gender and participation in different types of exercise. Females had lower self-esteem and body area satisfaction scores but higher eating disorder proneness, neuroticism, appearance evaluation, and overweight preoccupation scores than males. These findings add to the current literature focusing on the psychosocial and behavioral mechanisms associated with physical activity and exercise environments. The findings provide an effective supplement to promote public health-focused fitness programs and leisure-time physical activity motivation strategies.

## 1. Introduction

Physical activities and exercise programs during leisure time are important to promoting quality of life and overall well-being [1,2,3]. Research shows that regular leisure-time physical activity can help in the prevention or management of many chronic physical conditions [4,5]. It can also improve social, emotional, and mental well-being by reducing depression, stress, and social physique anxiety [6,7,8,9].

Today, physical inactivity is a significant problem that has serious consequences for individual and public health [10,11]. Therefore, it is important that people are motivated to participate in physical activities and exercise during their leisure time. Motivation can stem from fulfilling psychological needs such as autonomy, competence, and relatedness to health and psychological well-being factors, weight loss, and physical appearance [12,13,14,15,16,17].

The environment, whether natural or built, significantly influences the benefits, attitudes, preferences, and behaviors associated with exercise [5,18,19,20,21]. Exposure to natural and outdoor environments during exercise can enhance mood, increase happiness, reduce stress, and lead to higher levels of enjoyment, vitality, and positive engagement [3,20,21,22,23,24,25]. In fact, outdoor exercise has been associated with better psychological outcomes compared to indoor sessions [3,19,26,27,28,29,30].

Different forms of physical exercise are linked to different environments. For instance, gym-based activities tend to take place indoors or in constructed environments, while activities such as jogging, running, or cycling tend to take place in outdoor urban or natural environments [5]. Time spent exercising indoors has been shown to have a stronger connection with specific psychosocial factors, such as negative body image and eating disorders, than outdoor exercise [31]. Furthermore, the level of social interaction among outdoor and indoor exercisers is only a significant predictor of future exercise intentions for those who exercise outdoors [32].

People tend to limit themselves to exercises that feel safe, that they are comfortable performing, and that are within their capability levels [33]. The BIG 5 personality traits—agreeableness, extroversion, conscientiousness, neuroticism, and openness—can influence exercise choices [34,35,36]. Previous studies mentioned mixed results between personality traits and sports preferences, but it is generally agreed that personality impacts behavior and experiences during leisure activities [37,38].

The correlation between personality traits and physical workout involves factors like motivation, obstacles to exercise, choice of training, and workout venue preferences. Personality traits can influence these factors, shaping an individual’s engagement in physical activity [39,40]. Studies have shown that personality traits, motivations, and behavior regulation play a role in engagement in various types of physical activity. These factors correspond to distinct modes of physical activity engagement, and individuals exhibit different motivational dimensions and activity preferences. Therefore, interventions promoting physical activity enjoyment should be tailored to individual needs and personality traits [41,42].

The BIG 5 personality traits significantly impact the experience of physical activity enjoyment. For instance, extraversion moderates the correlation between social connectedness and joy, while conscientiousness and neuroticism moderate the association between perceived competence and joy [43,44]. Body image, a psychological construct encompassing cognitive, affective, and behavioral dimensions, can be predicted by personality traits [45]. Negative body image can impede exercise motivation and progress and affect the preferred workout location [46].

According to objectification theory [47], negative body image arises from internalizing societal beauty standards and viewing one’s body as an object for others to evaluate. This can influence exercise frequency and location preference. However, it does not prevent non-exercisers from starting to exercise. Gym workouts are more strongly linked to body image and eating disorders than outdoor exercise [46]. The duration of exercise at a gym exhibits a stronger correlation with body image and eating disorders in comparison to exercising outside a gym environment [31].

Appearance-based motivation to exercise [48] can moderate these associations. Exercise frequency is associated with a more positive body image, but this is moderated by high appearance motivation. A positive body image [49] is linked to factors like body functionality appreciation, acceptance by trainers and teammates, and high levels of intuitive eating. A negative body image can hinder participation and is associated with lower physical activity levels. Conversely, higher physical activity levels are linked to a positive body image [50]. High neuroticism and low conscientiousness predict male body dissatisfaction, while high neuroticism predicts it in females beyond the impact of BMI. Notably, personality traits do not significantly correlate with disordered eating scores in men, unlike in women [51].

Research suggests that the environment in which exercise takes place, whether indoor or outdoor, can significantly impact mental well-being. Outdoor exercise is often associated with greater psychological benefits, while indoor exercise, particularly in gyms, is more closely linked to eating disorders and negative body image [19,31,52,53].

The significance of this study is to provide a deeper understanding of the psychological impacts of different exercise environments to update the design of fitness programs according to individual needs, ultimately promoting leisure-time physical activity, healthier exercise habits, and improved mental well-being.

The aim of this study was to investigate the differences in eating attitudes, self-esteem, personality traits, and body image among experienced participants in leisure-time indoor and outdoor fitness programs. The hypotheses were as follows:(a)Participants in indoor fitness programs exhibit higher eating attitudes and lower self-esteem scores than outdoor participants.(b)Participants in indoor fitness programs show higher levels of neuroticism and lower levels of extraversion than outdoor participants.(c)Outdoor participants have higher positive body image scores than indoor participants.(d)Gender is a differentiating factor that influences eating attitudes, self-esteem, personality, and body image.

## 2. Materials and Methods

### 2.1. Participants

Participants included 1747 adults aged 34 ± 9.2 years (21–55 years old), of which 882 (50.5%) were men and 865 (49.5%) were women. These were divided into experimental and control groups. The experimental group consisted of 792 individuals who participate in regular (2+ years) outdoor physical fitness activities (running, cycling, trail running) and 735 who participate in regular (2+ years) indoor activities (cross-training, spinning, treadmill running, dance aerobics). The control group consisted of 220 individuals who did not participate in any physical activity or fitness program. The participants were selected through convenience sampling. The researchers published web-based surveys and sent questionnaire links to group managers within the target demographic. Participants were asked to complete the surveys voluntarily from their home computers. Forty-five questionnaires with missing data were excluded from the study. The study was approved by the Internal Ethics Committee (IEC) of the Department of Physical Education and Sport Science (DPESS), University of Thessaly, Greece.

### 2.2. Instrumentation

The participants completed the following: (a) a questionnaire on physical and demographic characteristics constructed for the needs of the study, (b) the 34-item version of the Multidimensional Body-Self Relations Questionnaire Appearance Scales (MBSRQ-AS) [54,55], (c) the Rosenberg’s Self-Esteem Scale (RSES) [56,57], the Eating Attitudes Test 26 (EAT-26) [58] and the Traits Personality Questionnaire 5 (TPQue5) [59,60], which is an abbreviated version of the Revised NEO Personality Inventory (NEO-PI-R) [61].

Physical and Demographic Characteristics. This study obtained data on the participants’ gender, age, height, and weight. The body mass index (BMI) was calculated using the participants’ self-reported weights and heights. Additionally, the participants provided information on their preferred exercise type, duration, frequency, and timing. Finally, the participants were asked to identify their primary motivation for engaging in physical activity.

Multidimensional Body–Self Relations Questionnaire–Appearance Scales (MBSRQ-AS). The MBSRQ-AS [55] consists of 34 items that assess aspects of body image related to physical appearance. The questionnaire includes the following scales: appearance evaluation (7 items, e.g., “I consider my body to be attractive”), appearance orientation (12 items, e.g., “I always check my appearance in a mirror whenever I can”), body areas satisfaction (9 items, e.g., “Face–features, face color”), preoccupation with being overweight (4 items, e.g., “I am aware of even the smallest changes that occur in my body”), and self-classified weight (2 items, e.g., “I think most people look at me and assume that I’m: (1) Much below normal weight, (2) Probably below normal weight, (3) Normal weight, (4) Probably above normal weight, (5) Much above normal weight. The items are rated on a 5-point Likert scale. Most items measure agreement (1 = strongly disagree to 5 = strongly agree), satisfaction (1 = very dissatisfied to 5 = very satisfied), or frequency (1 = never to 5 = very frequent). The self-rated weight scale has five specific responses (1 = very underweight to 5 = very overweight). The questionnaire has statistically significant validity and reliability for use in the Greek population [62].

Rosenberg’s Self-Esteem Scale. The RSES [57] is a 10-item scale designed to represent a continuum of statements about self-worth. Recent literature has distinguished between ‘state’ and ‘trait’ forms of self-esteem. The original RSES was designed to assess an individual’s global trait self-esteem and is scored using four response options, ranging from strongly agree to strongly disagree, and includes five positively worded items (e.g., “In general I am satisfied with myself”) and five negatively worded items (e.g., “I feel worthless sometimes”). The higher the score, the higher the individual’s self-esteem. The questionnaire showed statistically significant validity and reliability when used in the Greek population [63,64].

Eating Attitudes Test 26. The EAT-26 [58] is a self-reported questionnaire consisting of 26 items (e.g., “I think very often that I want to become thinner–I feel uncomfortable after I eat sweets”). These are presented on 6-point forced-choice Likert scales ranging from 1 = never to 6 = always. Responses to items 1–25 are scored on a 4-point scale (3 = always, 2 = usually, 1 = often, and 0 = sometimes, rarely, or never). Item 26 is reverse scored. The final score is calculated by summing items 1–26. The validity of the scale has been documented in many studies both at the international level and in Greece [58,65,66].

The Traits Personality Questionnaire 5. The TPQue5 [60] is a 101-item psychometric instrument consisting of 75 statements measuring the Big Five dimensions (neuroticism, extraversion, agreeableness, conscientiousness, and openness to experience) of personality (e.g., “When someone is in need, I always help him–I’m not a diffuse person in my events”). A further 26 statements assess social desirability. Responses are given on a 5-point Likert scale ranging from ‘completely disagree’ to ‘completely agree’ and are intended for men and women aged 17 years and over. The TPQue5 factor scales show excellent internal consistency and good test-retest reliability for the Greek population.

### 2.3. Procedure

The survey was conducted online between 1 January 2023 and 1 September 2023, with voluntary participation. The questionnaires were posted on practitioners’ websites via Google Forms. Participants were informed that their responses would remain anonymous and, to aid in completion, written explanations accompanied the questionnaires.

### 2.4. Data Analysis

All statistical analyses were performed in IBM SPSS Statistics ver. 29.0 (IBM Co., Ltd., Armonk, NY, USA). The Kolmogorov–Smirnov test was used to assess data distribution, and the samples obtained were found to be evenly distributed. Cronbach’s Alpha was used to measure the internal consistency of the scales (see Section 3.3). Physical attributes, including age, height, weight, and body mass index, were subjected to descriptive analysis. Additionally, the study examined exercise factors, such as years of experience, duration, and frequency of the time spent exercising per week, and initial motivations to engage in physical activity. A one-way analysis of variance (ANOVA) was conducted to investigate the differences in personality traits, body image, self-esteem, and eating attitudes between experienced indoor and outdoor participants. A two-way ANOVA was conducted to test the interaction between an exercise location (indoor vs. outdoor) and gender. Values for eta-squared (η^2^) and partial eta-squared (η*_p_*^2^) were also calculated to provide an indication of effect size. The threshold for statistical significance was set at *p* < 0.05.

## 3. Results

### 3.1. Descriptive Statistics and Frequencies

Outdoor exercisers averaged 6.1 ± 2.2 years of regular exercise, while indoor exercisers averaged 5.9 ± 2.1 years of regular exercise. The descriptive statistics are presented in Figure 1.

### 3.2. Somatometric Characteristics

The somatometric characteristics of the participants are shown in Table 1 and Table 2 as means (M) and standard deviations (SD) for the age, height, weight, and body mass index variables for the total sample and for the groups OE (outdoor exercisers), IE (indoor exercisers), and CG (control group).

### 3.3. Reliability Analysis

Cronbach’s alpha coefficient was calculated as a measure of the internal consistency of the scales. The internal consistency for the five factors of the TPQue5 questionnaire was quite high, ranging between 0.71 and 0.84. Specifically, Cronbach’s alpha consistency coefficient gave a value of *a* = 0.78 for extraversion, *a* = 0.84 for neuroticism, *a* = 0.71 for openness, *a* = 0.72 for agreeableness, and *a* = 0.75 for conscientiousness. For the MBSRQ-AS, the internal consistency of the five factors was quite high, ranging between 0.71 and 0.83. Specifically, Cronbach’s alpha consistency coefficient gave a value of *a* = 0.83 for appearance evaluation, *a* = 0.82 for appearance orientation, *a* = 0.81 for body area satisfaction (BASS), *a* = 0.71 for overweight preoccupation, and *a* = 0.83 for self-classified weight. The internal consistency for the RSES was *a* = 0.84, and for the EAT-26, *a* = 0.85. The above values of Cronbach’s *α* were considered satisfactory.

### 3.4. Analysis of Variance

Table 3 shows the mean values for all descriptive variables for each group. There were statistically significant differences between the three groups in all the questionnaires, as determined by the one-way ANOVA. Notably, the results indicated statistically significant differences (*F*_(2,1744)_ = 20.477, *p* = 0.001, η^2^ = 0.023) between the three groups on the EAT-26 questionnaire, which measures tendency or predisposition towards eating disorders. A post hoc Bonferroni analysis revealed a significant difference between the mean EAT-26 scores of the OE and IE groups, with the OE group demonstrating a lower score (*p* < 0.001). Likewise, the IE group had a notably different mean EAT-26 score than the CG (*p* = 0.002), with the CG displaying a lower score. No statistically significant difference was found in the mean EAT-26 scores between the OE and CG groups (*p* ≥ 0.5).

A one-way ANOVA showed statistically significant differences between the three groups (*F*_(2,1744)_ = 26.370, *p* = 0.001, η^2^ = 0.029) for RSES, which assesses global self-esteem. The post hoc Bonferroni test indicated that the OE group had significantly a higher mean self-esteem compared to both the IE (*p* < 0.001) and CG (*p* < 0.001) groups based on a one-way ANOVA. There was no significant difference in the self-esteem levels between the IE group and the CG (*p* ≥ 0.5). The TPQue5 questionnaire assesses personality traits using a one-way ANOVA, which revealed significant differences in the mean values for the three groups. Specifically, significant differences were found in extraversion between the groups (*F*_(2,1744)_ = 9.747, *p* = 0.001 η^2^ = 0.011). The OE group demonstrated a notably higher mean score for extraversion than either the IE group (*p* < 0.001) or the CG (*p* = 0.014), as determined by a post hoc Bonferroni test. There were no significant differences in the mean scores for extraversion between the IE group and the CG (*p* ≥ 0.5). However, for neuroticism, there were substantial differences between the mean scores of all three groups (*F*_(2,1744)_ = 33.351, *p* = 0.001 η^2^ = 0.037), as evidenced by the ANOVA. Notably, the OE group had a markedly lower mean score for neuroticism than either the IE group (*p* < 0.001) or the CG (*p* < 0.001). Additionally, the average neuroticism score of the IE group was significantly less than that of the CG (*p* < 0.001). The ANOVA found no statistically significant differences in the mean scores between the three groups for the personality trait of openness (*F*_(2,1744)_ = 1.863, *p* ≥ 0.5 η^2^ = 0.002). However, the results of the analysis revealed statistically significant differences between the three groups for agreeableness (*F*_(2,1744)_ = 5.888, *p* = 0.003 η^2^ = 0.007). The results of the post hoc Bonferroni test indicate that the OE group scored significantly higher for agreeableness compared to the IE group (*p* = 0.002). No statistically significant difference in the mean scores for agreeableness was found between the IE group and the CG (*p* ≥ 0.5). Statistically significant differences in the mean scores of the three respondent groups were observed for conscientiousness (*F*_(2,1744)_ = 14.845, *p* = 0.001 η^2^ = 0.017). According to the post hoc Bonferroni test, the OE group demonstrated a significantly higher mean for conscientiousness compared to either the IE group (*p* < 0.001) or the CG (*p* < 0.001). No significant difference was found in the mean conscientiousness scores of the IE and CG groups (*p* ≥ 0.5). The MBSRQ-AS questionnaire evaluates appearance-related aspects of body image construction. A one-way ANOVA revealed significant differences in the mean scores among the three groups. In particular, the results indicated a statistically significant variance (*F*_(2,1744)_ = 29.419, *p* = 0.001 η^2^ = 0.033) in the appearance evaluation factor. The post hoc Bonferroni analysis illustrated that the average score of the OE group significantly surpassed that of both the IE group (*p* < 0.001) and the CG (*p* < 0.001). Furthermore, the average score of the IE group exhibited a statistically significant increase compared to the CG (*p* < 0.001). ANOVA indicated significant differences between the three groups for appearance orientation, which assesses the degree of emphasis an individual places on their appearance (*F*_(2,1744)_ = 35.900, *p* = 0.001 η^2^ = 0.040). Significant differences were observed in the mean scores of the OE group compared to both the IE group (*p* < 0.001) and the CG (*p* = 0.003). Specifically, the mean score of the OE group was markedly lower than that of the other two groups. No significant differences were found between the IE and CG groups (*p* = 0.053). The ANOVA showed a statistically significant difference between the means of the three groups for the body areas satisfaction subscale (*F*_(2,1744)_ = 15.272, *p* = 0.001 η^2^ = 0.017). The OE group recorded the highest mean score and differed significantly from the CG (*p* < 0.001). The IE group obtained the second-highest mean score and was also significantly different from the CG group (*p* < 0.001). Statistically significant differences were found between the means of the three groups on the overweight preoccupation subscale (*F*_(2,1744)_ = 18.349, *p* = 0.001 η^2^ = 0.021). The OE group had the lowest mean and differed significantly from both the IE group (*p* < 0.001) and the CG (*p* < 0.001). However, the IE and CG groups did not differ significantly from each other. For the self-classified weight subscale, analysis of the results showed statistically significant differences between the means of the three groups (*F*_(2,1744)_ = 41.923, *p* = 0.001 η^2^ = 0.049). The OE group obtained the lowest mean score, which was significantly different from both the IE group (*p* < 0.001) and CG (*p* < 0.001). Additionally, there were significant differences between the IE and the CG (*p* < 0.001) groups, and the IE group had the second-lowest mean score on this subscale.

### 3.5. Multivariate Analyses

A two-way ANOVA was conducted to investigate the interplay between gender and exercise groups in the three experimental groups. The results revealed a statistically significant interaction effect on the EAT-26 scale (*F*_(2,1741)_ = 9.925, *p* = 0.001 η*_p_*^2^ = 0.011), self-esteem (*F*_(2,1741)_ = 5.008, *p* = 0.007 η*_p_*^2^ = 0.006), neuroticism (*F*_(2,1741)_ = 9.925, *p* = 0.001 η*_p_*^2^ = 0.008), and the MBSRQ-AS factors of appearance evaluation (*F*_(2,1741)_ = 3.035, *p* = 0.048 η*_p_*^2^ = 0.003), body areas satisfaction (*F*_(2,1741)_ = 6.539, *p* = 0.001 η*_p_*^2^ = 0.007), and overweight preoccupation (*F*_(2,1741)_ = 7.160, *p* = 0.001 η*_p_*^2^ = 0.008). In the EAT-26 assessment, there was a gender and exercise interaction specifically among the IE group (*p* < 0.001). Women were found to have significantly higher average scores, indicating a greater likelihood of developing eating disorders than men. Male participants in both the OE (*p* = 0.035) and IE (*p* < 0.001) groups, as well as in the CG (*p* < 0.001), demonstrated a significantly higher mean RSES compared to their female counterparts with regards to the relationship between gender and exercise impact on self-esteem. A significant interaction between genders was observed only among the OE group (*p* < 0.001), with women displaying a higher mean score for neuroticism. In the MBSRQ-AS questionnaire, factors examining body image pertaining to appearance demonstrated significant interactions. Specifically, the appearance evaluation factor exhibited a significant interaction only in the CG (*p* = 0.050). The BASS factor displayed a significant interaction in both the IE group (*p* = 0.025) and the CG (*p* = 0.019). Additionally, a significant interaction was observed in the IE group (*p* < 0.001) for overweight preoccupation. Figure 2A–F displays the mean values for all significant interactions between gender and the exercise location group.

## 4. Discussion

Modern society is motivated by health and well-being to participate in enjoyable physical fitness activities. The exercise environment can be a great way to enhance mood while improving lifestyle habits. The purpose of this study was to investigate the differences in eating attitudes, self-esteem, personality traits, and body image among experienced participants in leisure-time indoor and outdoor fitness activities.

The results confirmed both the first and second hypotheses concerning differences in eating attitudes, self-esteem, and personality traits. Outdoor exercisers were found to have lower scores for eating attitudes, higher scores for self-esteem, lower scores for neuroticism, and higher scores for extraversion, openness, agreeableness, and conscientiousness than indoor exercisers. The results are supported by previous research, which has shown that the built and natural environments in which human activities take place are key determinants of health and well-being [20,27]. There is considerable research evidence that exposure to natural and outdoor environments has significant psychological benefits, including reduced stress [21], improved overall well-being [24,25], and restored and enhanced mood [22]. Combining physical activity with direct exposure to nature may enhance mood, as both regular exercise and outdoor green environments can increase happiness and reduce stress [20,23,27,67]. Noseworthy et al. [5] found that outdoor exercise is associated with higher levels of enjoyment, vitality, positive engagement, and intention to return to exercise than indoor exercise. Lu and Hu [68] found that extraversion is significantly correlated with almost all types of leisure engagement, whereas neuroticism is unrelated to leisure engagement. The same researchers also found that extraversion is significantly positively correlated with leisure satisfaction, whereas neuroticism is significantly negatively correlated with leisure satisfaction and, while extraversion and neuroticism emerged as notable predictors of happiness, leisure satisfaction showed additional effects after controlling for personality traits and other domains of satisfaction.

The results for the third hypothesis on body image supported the assumption that individuals who engage in outdoor exercise have a more positive body image than those who exercise indoors. In addition, we found that outdoor exercisers scored higher on the evaluation of their appearance and the appreciation of individual body regions and lower on appearance orientation and concern about weight gain. As expected, indoor exercisers scored higher on the MBSRQ-AS than outdoor exercisers for appearance orientation and lower for appearance evaluation. This indicates that indoor exercisers are more concerned with their appearance and less satisfied with their body image than outdoor exercisers. Although the results of the study clearly show that people who exercise indoors are significantly more motivated by extrinsic goals, such as losing weight and improving their appearance, this did not seem to affect the scores for disordered eating and low self-esteem. Indoor exercisers did not score highly on the EAT-26 despite having lower scores on the RSES than the outdoor exercise group. Similarly, self-esteem did not seem to be affected by exercise location or motivation. Despite having lower scores than the outdoor exercise group, the indoor exercise group still scored high for self-esteem. However, for the personality trait of neuroticism, which has been shown to be a predictor of negative body image [51,69], both the indoor and outdoor exercise groups scored significantly lower than the control group. This might suggest that regular exercise, regardless of location, can help reduce neuroticism and its negative effects on body image. This is consistent with some studies that have found that exercise can buffer the negative effects of neuroticism on body image, self-esteem, and mood [70,71]. Regular exercise, regardless of location, gender, body size, or demographic differences, has been shown to promote the development of a positive body image and to have a protective effect. This finding is supported by three meta-analytic studies that examined the effects of exercise interventions on body image [72,73,74]. However, these studies included a variety of tools and instruments to measure body image parameters, different types, environments, and intensities of exercise, and diverse populations of participants. For instance, research by Cash [75] and Ginis et al. [76] found that changes in muscularity, body weight, and physical ability can have a positive effect on body image by increasing self-confidence, self-esteem, and body satisfaction. It is thought that these aesthetic changes may lead to improvements in body image independent of the objective physical changes caused by exercise [77]. In the sample in the present study, it is likely that such aesthetic changes in the bodies of the exercisers will occur, as they have long and stable involvement with exercise and all have normal BMIs.

For the fourth hypothesis, the results showed a significant interaction between the three groups and gender for eating attitudes, self-esteem, and neuroticism. There was also a significant interaction for the MBSRQ-AS subscales of appearance orientation, BASS, and overweight preoccupation. This is consistent with many studies that have shown that gender is a differentiating factor that influences personality, self-esteem, and body image. For instance, women tend to score higher for neuroticism and lower for self-esteem than men and are more likely to experience body dissatisfaction and disordered eating than men [78,79,80,81,82]. The factor of exercise participation has been shown to have different effects on men and women with regard to personality, self-esteem, and body image [83]. As discussed earlier, exercise participation is considered an important variable when explaining gender differences in personality, self-esteem, and body image [84,85]. Exercise plays an important role in developing and controlling weight in both men and women, which may influence body image perceptions and eating attitudes. The goals for exercising that motivate participants may be determinants of unhealthy eating behaviors, such as bingeing, purging, or restricting [86]. In line with previous studies, the present research found that women have different reasons for participating in exercise programs than men [87,88]. Specifically, women tend to exercise more for appearance and weight-related goals, while men tend to exercise more for health and enjoyment-related goals. As a result, women are more likely to have higher levels of body dissatisfaction, body image concerns, and disordered eating than men [89,90,91]. This present study found a significant interaction between gender and physical activity for eating attitudes (EAT-26). This was particularly true for those who exercised indoors, with women having significantly higher mean scores than men [86]. However, their overall scores did not indicate a potential risk for eating disorders, suggesting that indoor exercise may not have a negative impact on women’s eating attitudes. In addition, the study found a significant interaction between gender and exercise on self-esteem, which was measured by the RSES. Male participants in all three groups showed significantly higher self-esteem than the female participants. This suggests that gender may have a stronger influence on self-esteem than exercise location or motivation. Incidentally, despite the statistical differences between the sexes, the RSES scores showed that women tended to have high self-esteem [86,92]. There appeared to be a significant interaction between outdoor exercisers for the personality trait of neuroticism. However, women in this group had lower mean neuroticism scores than the other two groups. Although the interaction between exercise groups and gender is significant for eating attitudes, self-esteem, and neuroticism, these factors do not seem to have a negative impact on the psychological health of the participants. Regular exercise over a few years seemed to play a more important role in the overall healthy psychological well-being of the female participants [93,94]. This is because it may bring benefits such as a healthy BMI and positive body image for the exercising women [73]. In fact, outdoor exercisers tend to have more ‘athletic’ lifestyles, which may explain why they have lower levels of neuroticism and higher levels of self-esteem and body image than indoor exercisers. They not only pay attention to their diet, but also use various nutritional supplements to maximize their sports performance. This attitude shows that appearance is not prioritized, and the body becomes the means to achieving recreational exercise goals.

This study has some limitations. Although the study included a large sample, it is crucial to acknowledge that convenience sampling may not accurately represent all of the outdoor and indoor physical activity participants. Consequently, we advise that the results be interpreted with caution. The motivations behind exercise participation, which can influence food intake disorders, were not thoroughly explored. The questionnaire lacked detail on the reasons for participating in exercise. The voluntary nature of survey participation may have skewed the results toward those with a high agreeableness personality trait. The sample was limited to certain sports, restricting the generalizability of the findings to all indoor and outdoor exercisers. Other variables, such as physical activity levels, exercise intensity, environmental factors, and ambient temperature factors, were not controlled for. Therefore, future research should address these limitations by using larger and more diverse samples, objective measures, and experimental designs.

The findings of this study could have several implications for the field of leisure-time physical exercise and the promotion of mental well-being and public health. By comparing the personality traits and body image of outdoor and indoor exercisers, we can gain insight into the factors that influence their choice of exercise environment. To encourage exercise and capture its positive benefits, appropriate exercise programs can be designed according to an individual’s personality and body image characteristics. Furthermore, the study shows that exercise location and motivation affect body image, eating attitudes, self-esteem, and personality traits in different demographics. This underlines the need for tailored exercise regimes to promote mental well-being. Regular exercise can improve a person’s body image and psychological health, regardless of location or motivation, suggesting its preventative and remedial potential for various mental health conditions. Collaborations with leisure and mental health professionals could help with the development of strategies to promote regular physical activity. Additionally, outdoor physical exercise participants tend to lead more active lifestyles due to their focus on nutrition and supplements, which could enhance their body image and psychological health. The impact of different diets and supplements on these factors could be further investigated.

## 5. Conclusions

The present research indicated that indoor and outdoor exercise and motivation have different effects on the body image, eating attitudes, self-esteem, and personality traits of experienced exercisers. This study further showed that consistent exercise for several years could have beneficial impacts on these psychological aspects, irrespective of the exercise setting and motivation. Finally, this research indicated that active individuals who participate in outdoor sports have a more athletic lifestyle, positively affecting their attitudes toward body image and exercise. These results are significant to the field of leisure, exercise psychology, and the public health sector, providing valuable perspectives for policymakers and practitioners to establish exercise during leisure time as a preventive and therapeutic tool for various health difficulties and concerns.

## Figures and Tables

**Figure 1 healthcare-12-00047-f001:**
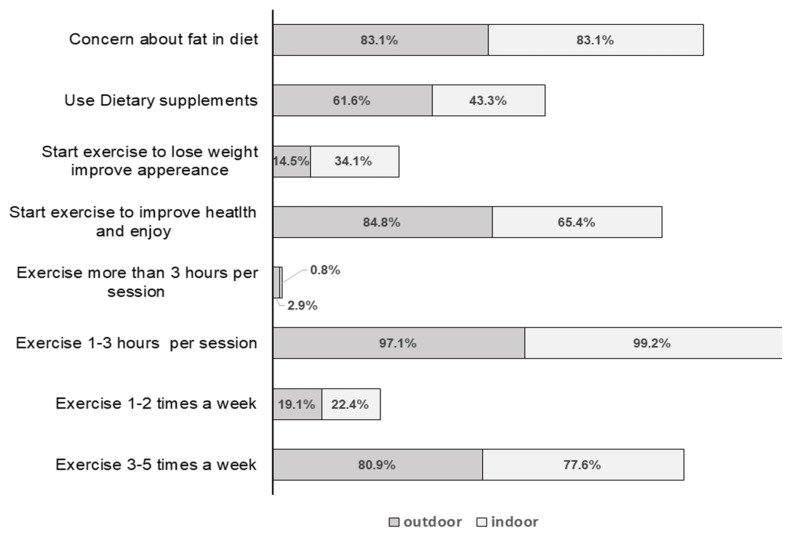
Percentages of participants in indoor and outdoor fitness activities in exercise duration and frequency, attitudes towards diet, supplements, and exercise expectations.

**Figure 2 healthcare-12-00047-f002:**
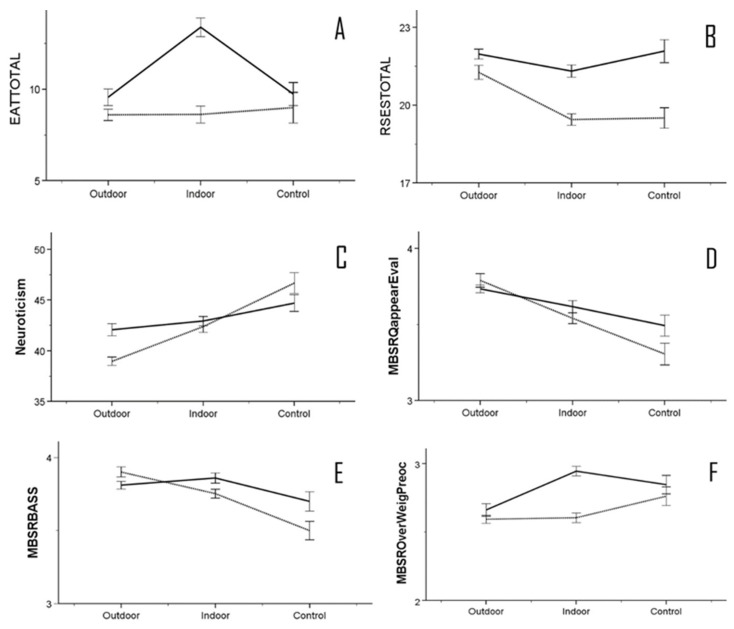
Interactions between gender and exercise location group. Solid line = men, dotted line = women. (**A**) EAT-26, (**B**) RSES self-esteem, (**C**) TPQue5 neuroticism, (**D**) MBSRQ appearance evaluation, (**E**) MBSRQ BASS, and (**F**) MBSRQ overweight preoccupation.

**Table 1 healthcare-12-00047-t001:** Somatometric characteristics in the entire sample and by sample group.

		Age	Height	Weight	BMI
		M ± SD	M ± SD	M ± SD	M ± SD
Total	TS N = 1747	33.9 ± 9.20	1.74 ± 0.081	70.7 ± 12.8	23.2 ± 3.07
	OE N = 792	36.0 ± 8.96	1.76 ± 0.77	72.6 ± 11.8	23.3 ± 2.74
Groups	IE N = 735	31.8 ± 8.69	1.74 ± 0.78	68.5 ± 13.0	22.9 ± 3.23
	CG N = 220	33.5 ± 10.0	1.72 ± 0.82	71.2 ± 14.4	24.0 ± 3.45

TS = total sample, OE = outdoor exercisers, IE = indoor exercisers, CG = control group, N = number of people in the sample per group, M = mean, SD = standard deviation, Age = years, Height = meters, Weight = kg, BMI = body mass index.

**Table 2 healthcare-12-00047-t002:** Somatometric characteristics of the male and female participants by sample group.

	Groups			Age	Height	Weight	BMI
			N	M ± SD	M ± SD	M ± SD	M ± SD
	M	OE	533	37.8 ± 8.84	1.80 ± 0.058	78.9 ± 7.81	24.4 ± 2.14
	F	OE	259	32.5 ± 8.12	1.68 ± 0.054	59.6 ± 7.23	21.0 ± 2.40
Gender	M	IE	262	33.8 ± 9.29	1.80 ± 0.061	81.1 ± 8.88	25.0 ± 2.43
	F	IE	473	30.7 ± 8.14	1.68 ± 0.050	61.5 ± 9.02	21.7 ± 3.03
	M	CG	87	35.3 ± 9.64	1.79 ± 0.061	83.8 ± 11.8	26.0 ± 3.08
	F	CG	133	32.2 ± 10.1	1.67 ± 0.052	63.1 ± 8.93	22.6 ± 2.99

M = male, F = female, OE = outdoor exercisers, IE = indoor exercisers, CG = control group, N = number of people in the sample per group, M = mean, SD = standard deviation, Age = years, Height = meters, Weight = kg, BMI = body mass index.

**Table 3 healthcare-12-00047-t003:** All questionnaire descriptive variables for each group.

Groups	OE	IE	CG
	N = 792	N = 735	N = 220
	M ± SD	M ± SD	M ± SD
EAT-26	8.91 ± 7.25 **	11.7 ± 10.2	9.44 ± 7.45 *
RSES	21.7 ± 4.36 **	20.1 ± 4.52	20.5 ± 4.57
TPQue5 Extraversion	52.7 ± 7.62 **	51.0 ± 7.87	51.0 ± 7.61
TPQue5 Neuroticism	40.0 ± 9.71	42.7 ± 9.68 **	45.5 ± 9.54 **
TPQue5 Openness	49.5 ± 7.57	48.8 ± 7.29	49.3 ± 8.07
TPQue5 Agreeableness	53.1 ± 6.55 *	52.0 ± 6.95	52.7 ± 7.00
TPQue5 Conscientiousness	51.5 ± 6.98 **	49.8 ± 7.71	49.0 ± 7.87
MBSRQ-AS AppearEval	3.75 ± 0.646 **	3.57 ± 0.722 **	3.38 ± 0.764
MBSRQ-AS AppearOrient	3.25 ± 0.637 **	3.51 ± 0.580	3.40 ± 573
MBSRQ-AS BASS	3.84 ± 0.596 **	3.79 ± 0.627 **	3.58 ± 0.693
MBSRQ-AS OverWeigPreoc	2.62 ± 0.692 **	2.82 ± 0.715	2.81 ± 0.728
MBSRQ-AS SelfPercWeig	2.98 ± 0.684 **	3.18 ± 0.710	3.45 ± 0.757

OE = outdoor exercisers, IE = indoor exercisers, CG = control group, N = number of people in the sample per group, M = mean, SD = standard deviation, AppearEval = appearance evaluation, AppearOrient = appearance orientation, BASS = body areas satisfaction, OverWeigPreoc = overweight preoccupation, SelfClasWeig = self-classified weight. * = statistically significant (*p* < 0.050), ** = statistically significant (*p* = 0.001).

## Data Availability

The data presented in this study are available on request from the corresponding author.

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
