# Peer review of "Investigating Differences in Personality Traits, Self-Esteem, Eating Attitudes, and Body Image among Participants in Indoor and Outdoor Fitness Activities"

_healthcare, 2023, doi:10.3390/healthcare12010047_

Round 1

Reviewer 1 Report

Comments and Suggestions for Authors

Thank you very much for letting me review this very interesting research. In my opinion the research has a correct design and the scientific method has been followed. As a general comment, the text is too long and some parts such as the introduction could be reduced.

Below are the specific comments.

Introdution

The introduction is well written and justifies the research but is very long. In some paragraphs the information could be summarized.

Line 44. A more relevant reference is necessary. Justify with the study "Exercise is the real polypill" by Fiuza-Luces and collaborators (2013) or similar.

Line 48. Add reference to justify

line 70. Add reference to justify

Line 137. Add reference to justify

Material and Methods

Include information about the ethics committee in participants section or procedures. Add access specific information to the sample.

Results

The information in the first paragraph could be summarized in a bar diagram.

On the other hand, four tables are not necessary to report the age, weight, height and BMI of the participants. It would be necessary to unify table 1 and 2, including one more row on the total in the second table. Table 3 and 4 can also be unified. In this way the length of the paper is reduced.

Table 5 does not show the significant differences between the groups. This would considerably reduce the information provided in the second paragraph.

Discussion

In the third paragraph of the discussion it is necessary to justify the statements made between lines 520 and 542.

Author Response

Thank you very much for the time and effort that you dedicated to providing feedback on our manuscript. We are grateful for the insightful comments and valuable improvements to our paper. We have incorporated most of the suggestions. Those changes are highlighted in red within the manuscript. Please see below for a point-by-point response to your comments and concerns.

  1. Comments from Reviewer “As a general comment, the text is too long and some parts such as the introduction could be reduced”. “The introduction is well written and justifies the research but is very long. In some paragraphs the information could be summarized”.

Author Response: We reduced the words and changed the length of the text especially the abstract and introduction section.

  1. Comment from Reviewer “Line 44. A more relevant reference is necessary. Justify with the study "Exercise is the real polypill" by Fiuza-Luces and collaborators (2013) or similar.”

Author Response: We add the relevant reference in the introduction part, L38, L541

  1. Comments from Reviewer “Line 48. Add reference to justify, “line 70. Add reference to justify”, “Line 137. Add reference to justify”.

Author Response: We change completely the introduction part according to your suggestions.

  1. Comment from Reviewer “Include information about the ethics committee in participants section or procedures. Add access specific information to the sample”.

Author Response: We add the information in the manuscript, L131-133

  1. Comment from Reviewer “ The information in the first paragraph could be summarized in a bar diagram.

Author Response: We add the diagram, L207

  1. Comment from Reviewer “On the other hand, four tables are not necessary to report the age, weight, height and BMI of the participants. It would be necessary to unify table 1 and 2, including one more row on the total in the second table. Table 3 and 4 can also be unified. In this way the length of the paper is reduced”.

Author Response: We made the changes that you suggest, L220-231.

  1. Comment from Reviewer “Table 5 does not show the significant differences between the groups”.

Author Response: We add the significant differences in Table 3, L320-325.

  1. Comment from Reviewer “Discussion. In the third paragraph of the discussion, it is necessary to justify the statements made between lines 520 and 542.”

Author Response: We add the references in L444, L451, L452

Reviewer 2 Report

Comments and Suggestions for Authors

The hypothesis of the work is in many ways quite obvious and the novelty of this part of the article is poorly visible. In general, the entire Introduction should be reworked, because it is written too long and very popular. Given that some of the surveyed people were engaged in outdoor sports activities, it is completely incomprehensible that the text does not mention and consider the influence of the ambient temperature factor – this aspect is not considered at all in the work. The work contains an analysis of gender differences, which is fundamental in matters of self-esteem. Perhaps it would be worth emphasizing this division right away, because for the most part the self-esteem of men and women is determined by different determinants - and you need to start right away, make two samples and compare classes on the street and in the gym inside them, because otherwise it looks like it was added artificially.

Line 444 discussion - why start a discussion by repeating the purpose of the study and the hypothesis – does it look bad? Some of the suggestions in the discussion are similar to those in the introduction (L455 — 456). The quote phrase (5) on Line 457-458 looks extremely unfortunate. In general, the novelty of the work looks rather unconvincing – and the conclusions are quite obvious. Why write about practical recommendations in the annotation to the article? (Line 30 — 34). In general, in the opinion of the reviewer, the article is not written in a completely scientific language and requires editorial editing. Special correction and editing is required in the introduction and discussion.

Author Response

Thank you very much for the time you dedicated to providing feedback on our manuscript. We are grateful for the valuable comments to our paper. We have incorporated most of the suggestions. Those changes are highlighted within the manuscript. Please see below for a point-by-point response to your comments and concerns.

  1. Comments from Reviewer “The hypothesis of the work is in many ways quite obvious and the novelty of this part of the article is poorly visible. In general, the entire Introduction should be reworked, because it is written too long and very popular”.

Author Response: We reduced the words and changed the length of the text  to improve the novelty. You can find the changes in L12-117

  1. Comment from Reviewer “Given that some of the surveyed people were engaged in outdoor sports activities, it is completely incomprehensible that the text does not mention and consider the influence of the ambient temperature factor – this aspect is not considered at all in the work.”

Author Response: As you rightly point out, we are not referring to environmental factors and temperatures. So we added it to the research limitations, L467

  1. Comments from Reviewer “The work contains an analysis of gender differences, which is fundamental in matters of self-esteem. Perhaps it would be worth emphasizing this division right away, because for the most part the self-esteem of men and women is determined by different determinants - and you need to start right away, make two samples and compare classes on the street and in the gym inside them, because otherwise it looks like it was added artificially.”

Author Response: We refer an analysis of gender differences, L423-457.

  1. Comment from Reviewer “Line 444 discussion - why start a discussion by repeating the purpose of the study and the hypothesis – does it look bad?”

Author Response: We made changes according to your suggestion, L356-358

  1. Comment from Reviewer “Some of the suggestions in the discussion are similar to those in the introduction (L455 — 456).”

Author Response: We change the introduction section completely so there are no more similarities.

  1. Comment from Reviewer “The quote phrase (5) on Line 457-458 looks extremely unfortunate.”

Author Response: We deleted it.

  1. Comment from Reviewer “In general, the novelty of the work looks rather unconvincing – and the conclusions are quite obvious.”.

Author Response: We made significant changes throughout the manuscript to overcome the weaknesses you mention.

  1. Comment from Reviewer “Why write about practical recommendations in the annotation to the article? (Line 30 — 34).”

Author Response: We are usually asked to mention the practical  recommendations. However, we removed them towards the end of the discussion, L470-485.

9             Comment from Reviewer “In general, in the opinion of the reviewer, the article is not written in a completely scientific language and requires editorial editing. Special correction and editing is required in the introduction and discussion.”

Author Response: We made corrections and changes to improve the language. However, if there is still a problem, we will proceed with editing through journal's procedures.

Reviewer 3 Report

Comments and Suggestions for Authors

Dear authors,

Congrats for this contribution. Just few minor recommendations are presented:

a) Please describe hypotheses.

b) Please provide a better integration of the study aims when justifying the need of the study.

c) Sample items could be presented in the description of instruments.

Author Response

Thank you very much for the time and effort that you dedicated to providing feedback on our manuscript. We are grateful for the insightful comments and valuable improvements to our paper. We have incorporated all of your suggestions. Those changes are highlighted in red within the manuscript. Please see below for a point-by-point response to your comments and suggestions.

  1. Comments from Reviewer “Please describe hypotheses”.

Author Response: We described the hypotheses, L108-116.

  1. Comment from Reviewer “Please provide a better integration of the study aims when justifying the need of the study.”

Author Response: We add your suggestion, L102-105

  1. Comments from Reviewer “Sample items could be presented in the description of instruments”

Author Response: We add the sample items according to your suggestion, L149-153, L163-165, L169-170, L178-179.

Round 2

Reviewer 1 Report

Comments and Suggestions for Authors

Thanks you for aclarations. All comments has been revised.

Author Response

Thank you for providing many helpful comments to improve the paper. We appreciate your time and effort in this regard. 

Reviewer 2 Report

Comments and Suggestions for Authors

In general, the authors have largely reworked the text, which has become significantly more attractive. In general, the presented text is more suitable for publication in the journal. At the same time, it should be noted. in the opinion of the reviewer, the authors were unable to correct one of the most important remarks addressed to them - they did not provide materials on the relationship of the studied phenomenon - physical exercise and nutritional characteristics with ambient temperature - but they collected the material from January to September . This, in the opinion of the reviewer, is still a disadvantage of the study, but apparently initially the authors were not ready to consider and investigate this issue

Author Response

We really appreciate your valuable comments and thank you very much. We feel obliged to emphasize that the main purpose of the present study was not to examine the effect of the environment on nutrition characteristics and exercise, but the differences in personality, self-esteem, eating attitudes and body image. We emphasize that we examined eating attitudes with the questionnaire (EAT-26) which is purely a variable that detects the tendency of people to manifest eating disorders. We did not proceed with a thorough investigation of the nutrition habits. According to the literature, attitudes remain relatively stable throughout life and show the way in which the individual perceives social phenomena. The concept of attitude is a multidimensional concept consisting of the cognitive, affective and behavioral component of the individual. Attitudes are formed within the family and social environment. On the one hand they are influenced by social and psychological factors, which govern the social agencies that shape attitudes (family, friends, peers, media, school, etc.) and on the other hand by cognitive factors as the cognitive side of attitude refers mainly to the cognitive functions of the individual that characterize human intelligence. Eating attitudes are directly linked to disordered eating behaviors and food-related behaviors, as well as to disturbances in the thinking and feeling. In addition, they are associated with intense worry, disturbed attitude, and obsessive-compulsive disorder in relation to food, body weight, and body image. Finally, we would like to highlight that in our country (where the research was taken place) its geographical extent is limited. Greece is a Mediterranean country where there are no significant environmental differences, nor a significant difference in temperature by region. The temperature difference is of the order of 3-4 degrees Celsius per territory. Thus, according to the literature, it was considered by the researchers that it cannot significantly change the above factors.

However, in order not to confuse the reader with the word “habits”, we proceeded to replace it the word " attitudes both in the title and in the rest of the text. The changes have been highlighted with red in the manuscript.